# Validating Noisy Label Learning in Noisy Clinical Datasets

**André Aquilina**                                   ANDRE.AQUILINA@DYSISMEDICAL.COM

**Emmanouil Papagiannakis**                MANOLIS.PAPAGIANNAKIS@DYSISMEDICAL.COM

*DYSIS Medical Ltd, Edinburgh, UK*

**Editors:** Accepted for publication at MIDL 2024

## Abstract

Noise in ground truth labels limits model performance; in response, learning with noisy labels (LNL) has received much attention in recent years. However, most research has been applied to competition datasets where a clean (noiseless) test set is available. A gap exists in applying LNL to practical datasets such as colposcopy, where such clean sets are not available. By synthesizing additional noise, targeted to mimic real-world errors, to the training labels, and using an imperfect test set, we demonstrate that LNL methods outperform traditional learning, thus bridging this gap.

**Keywords:** colposcopy, learning with noisy labels, medical imaging, deep learning, cervical cancer, histology classification

## 1. Introduction

Colposcopy, the visual examination of the cervix uteri following an abnormal screening result, is crucial for cervical cancer prevention. Deep-learning (Piccialli et al., 2021) can reduce the subjectivity and enhance the performance of conventional visual colposcopy (Wentzensen et al., 2017). However, similarly to other medical imaging applications, high-quality ground truth datasets are scarce. Histology readings of biopsy samples, the ground truth gold standard, are in themselves subjective (Stoler et al., 2001), and furthermore, suffer from verification bias as only biopsied areas have a label, and negativity is rarely confirmed. Label noise can be significant, impacting model reliability.

Learning with Noisy Labels (LNL) (Song et al., 2022), a method to address this issue, typically requires a clean ground truth test set for confirmation, but this is rare in real-world datasets (Wei et al., 2021), particularly in colposcopy.

Our research introduces synthetic noise injection as a form of ablation study, providing practitioners with a mechanism to validate performance of noisy label learning models on real-world datasets that suffer from label noise. By increasing the noise in the training labels with the injection of (additional) synthetic noise, and using a comparatively cleaner test set for evaluation, we aim to provide a method to confirm LNL model effectiveness on real-world clinical datasets.

## 2. Methodology

### 2.1. Synthetic Noise Injection

We used a dataset of colposcopy images (N=4,948) (Aquilina and Papagiannakis, 2024; Perkins et al., 2022) with histology ground truth labels of Negative/Low-grade (N/LG) or High-grade (HG). The data suffers from inherent label noise of unknown quantity.

To reflect potential real-world noise, we introduce two types of additional label noise to the training process:

- Selective Verification Noise: Mimics verification bias by altering the labels of a random subset of High-grade findings to represent partial verification.

- Symmetrically Random Noise: Represents random labeling errors uniformly across both classes.

Noise is modeled as:

$$y = \begin{cases} y_j & \text{with probability } p_{miss,ij} \\ y_i & \text{otherwise} \end{cases} \tag{1}$$

where $y_j$ is a misclassified label, $y_i$ is the original label, and $p_{miss,ij}$ is the misclassification probability from class $y_i$ to $y_j$. For symmetrically random noise, $p_{miss,ij}$ is uniform across all classes; for selective verification noise, it varies, reflecting real-world verification bias where the likelihood of going from HG to N/LG is significantly higher than the contrary.

## 2.2. Clean Test Dataset Utilization

A subset of the data, inherently noisy, but devoid of synthetic noise, was held aside to allow us to evaluate the model's performance and assess the effectiveness of the LNL technique at different levels of synthesized noise in the training process.

## 3. Experiments and Results

### 3.1. Experimental Setup

The dataset was split at 80-10-10 for training, validation, and testing, and a 10-fold cross-validation was utilized for thorough evaluation.

**Data Preparation:** Synthetic noise, as described above, was injected into the training and validation sets to simulate real world conditions, at levels ranging from 0% to 50% for symmetric noise, and up to 90% for verification noise. In the latter, the probability of going from N/LG to HG was held at 0% throughout.

**Model Selection:** We compared a ResNet50 (He et al., 2016) and a DivideMix ResNet50 (Li et al., 2020) to assess noise handling capabilities. Hyperparameters were kept constant across models for meaningful analyses.

### 3.2. Evaluation Metric

We used the Area Under the Curve (AUC) of the receiver operating characteristic (ROC) curve as a global metric to assess model performance in binary image classification. Model selection was based on performance on the validation set.

### 3.3. Results

Model performance in the two experiments is summarized in Figure 1. Both models suffer performance degradation as noise injection is increased. However, the rate of degradation of the standard ResNet50 model far outpaces that of the DivideMix in both experiments. The

trend demonstrates the DivideMix framework's noise resilience, especially as it maintains about **95%** of its ROC AUC across noise levels in the verification noise experiment.

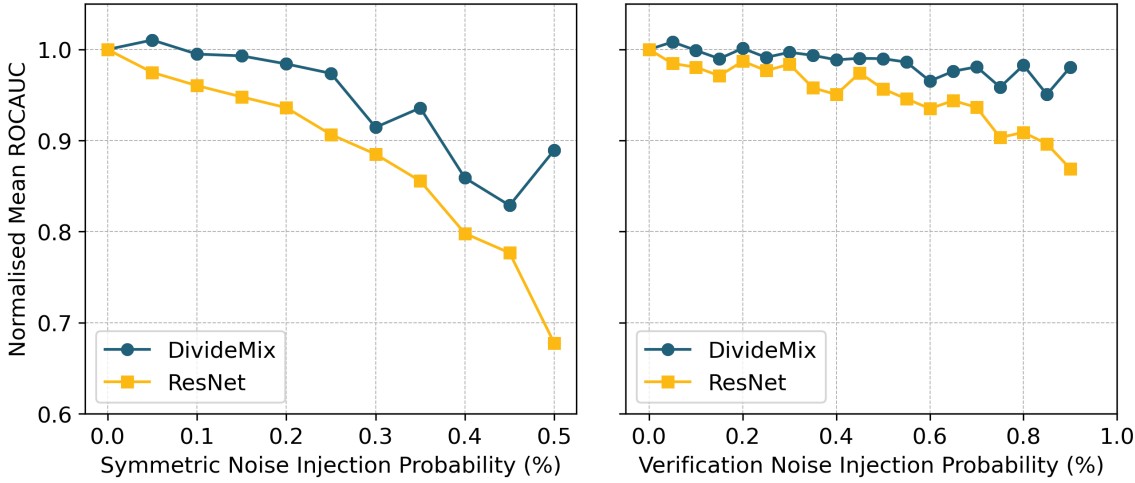

Figure 1: Degradation of model performance over various noise levels for ResNet50 and DivideMix. Performance was normalised to baseline ROCAUC of each model (0.7263 and 0.7125 respectively).

## 4. Discussion

Given its noise resilience and effectiveness across a range of noise levels compared to a standard ResNet50, as evidenced in Figure 1, DivideMix presents a viable solution for handling noisy labels in clinical datasets like ours. Performance in the verification experiment, where noise was injected in a fashion that we expect it to manifest in colposcopy datasets, further corroborates DivideMix's and more generally, LNL's, applicability to such real-world scenarios.

Our methodology, demonstrates a way to verify LNL techniques work in practice without the luxury of a clean test set, bridging the recognised gap between competition dataset performance and real world applicability (Wei et al., 2021).

This finding is particularly useful for practitioners, as it enables the productionization of noisy label learning models without the necessity of a perfectly clean gold standard test set. Such an approach significantly reduces the barrier to implementing LNL in practical, imperfect data scenarios commonly encountered in healthcare and beyond.

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
