# OpenReview forum: "Validating Noisy Label Learning in Noisy Clinical Datasets"
_MIDL.io/2024/Short_Papers — MIDL 2024 Short Papers_

### Official Review · Reviewer_A4sw · 2024-04-18

**Confidence:** 3
**Final Rating:** 3.5

**Review:**

The paper proposes to synthetically generate noisy (in the form of label noise) training cases in colposcopy, and evaluate using a noisy test set since clean test sets are not available.

Strengths:
-	Relatively unaddressed application (colposcopy)
-	Experiment with increasing the noise probability makes the results more convincing
-	Important to invesetigate model robustness

Weaknesses:
-	If the dataset is not public definitely more details need to be provided
-	No motivation / explanation of the DivideMix network which I don’t think is as well-known as the regular ResNet50
-	The conclusion of not neededing gold standard labels seems overstated. The results are interesting but I don’t think we can conclude from a higher AUC on an imperfect test test, that AUC would also be higher on a clean test set?

I think the paper could be potentially interesting to discuss at the conference if there is room, but there are also some weaknesses which is reflected in my score.

---

### Decision · Program_Chairs · 2024-04-26

Accept